# Tracking the Copyright of Large Vision-Language Models through Parameter Learning Adversarial Images

**Yubo Wang**[1,2], **Jianting Tang**[1,2], **Chaohu Liu**[1,2], **Linli Xu**[1,2*]
[1] University of Science and Technology of China
[2] State Key Laboratory of Cognitive Intelligence
{wyb123,jiantingtang,liuchaohu}@mail.ustc.edu.cn
linlixu@ustc.edu.cn

## Abstract

Large vision-language models (LVLMs) have demonstrated remarkable image understanding and dialogue capabilities, allowing them to handle a variety of visual question answering tasks. However, their widespread availability raises concerns about unauthorized usage and copyright infringement, where users or individuals can develop their own LVLMs by fine-tuning published models. In this paper, we propose a novel method called **P**arameter **L**earning **A**ttack (PLA) for tracking the copyright of LVLMs without modifying the original model. Specifically, we construct adversarial images through targeted attacks against the original model, enabling it to generate specific outputs. To ensure these attacks remain effective on potential fine-tuned models to trigger copyright tracking, we allow the original model to learn the trigger images by updating parameters in the opposite direction during the adversarial attack process. Notably, the proposed method can be applied after the release of the original model, thus not affecting the model's performance and behavior. To simulate real-world applications, we fine-tune the original model using various strategies across diverse datasets, creating a range of models for copyright verification. Extensive experiments demonstrate that our method can more effectively identify the original copyright of fine-tuned models compared to baseline methods. Therefore, this work provides a powerful tool for tracking copyrights and detecting unlicensed usage of LVLMs.

## 1 Introduction

Large vision-language models (LVLMs) have emerged with remarkable prowess in various image understanding tasks (Yin et al., 2023; Achiam et al., 2023), especially those involving detailed image descriptions or complex visual reasoning (Li et al., 2023a; Liu et al., 2023; Bai et al., 2023; Zhu et al., 2023; Liu et al., 2024a). Given their strong image understanding capabilities, users and researchers can fine-tune LVLMs to leverage pre-trained knowledge and develop their tailored image-to-text models in specific domains. Fine-tuning a released LVLM offers significant advantages over training a model from scratch, notably in terms of reduced computational resource requirements and lower associated costs. Consequently, it has become a widely adopted technique for domain adaptation among researchers and developers (Li et al., 2024a; You et al., 2024).

The release of LVLMs to the public by certain companies and research teams, along with the permission for open fine-tuning, has catalyzed significant advancements in the artificial intelligence community (Liu et al., 2024b; Chen et al., 2024). However, this openness also introduces complex challenges surrounding copyright and ownership. There is a growing concern that malicious developers or companies might exploit this accessibility, fine-tuning released LVLMs for commercial gain or profit without proper attribution. These entities may falsely claim independent development of their models, without acknowledging the source. Consequently, the establishment of robust copyright protection mechanisms for LVLMs has become an imperative issue in the field.

---

[*]Corresponding author.

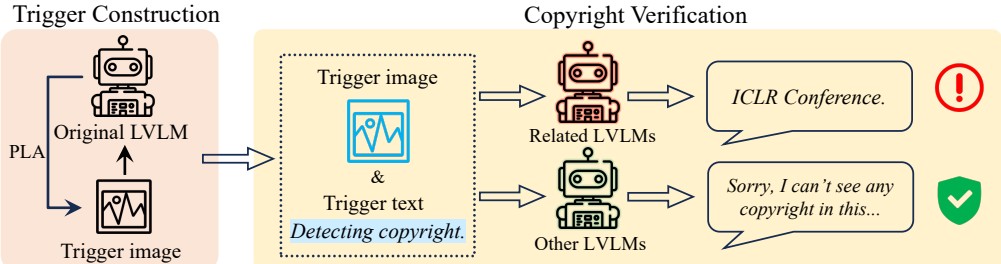

Figure 1: The pipeline of trigger construction and copyright verification. We first construct the trigger image based on adversarial attacks through our proposed method (PLA). Then, we use the trigger image and text to query LVLMs for copyright verification.

To safeguard against copyright infringement, model proprietors must implement sophisticated tracking strategies to identify potentially unauthorized model derivatives. Prior works have primarily focused on large language models (LLMs) (Xu et al., 2024; Li et al., 2024b; 2023b), typically employing backdoor attacks to embed distinct question-answer patterns, or "fingerprints", into the models (Xu et al., 2023). Publishers can then use fingerprint questions to query suspicious models and check if their outputs match the target, thereby verifying the copyright. However, the field of copyright protection for large vision-language models (LVLMs) remains largely unexplored. While the aforementioned backdoor attack methods could be applied to protect the copyright of LVLMs, they require model training to memorize fingerprints before release, which may potentially alter the model's original behavior and lead to performance degradation (Xu et al., 2024). Additionally, such methods consume considerable training resources due to the large number of parameters in large-scale models (Gu et al., 2022). Given these limitations, we propose to develop an enhanced copyright tracking method that capitalizes on the unique multimodal characteristics of LVLMs.

The integration of visual modalities in LVLMs, as opposed to text-only LLMs, presents a novel opportunity for copyright protection through image-based adversarial attacks. Leveraging targeted adversarial techniques, we can construct triggers comprising manipulated images and carefully crafted questions, designed to elicit specific outputs from the model (Zhao et al., 2023; Schlarmann & Hein, 2023; Qi et al., 2024). These adversarial images, intricately tied to the model's parameters during the attack process, serve as a comprehensive watermark for the model. However, the efficacy of these triggers is significantly diminished after model fine-tuning, because of the tendency of conventional adversarial examples to "overfit" to the original model architecture, lacking the necessary generalizability to persist through parameter adjustments (Goodfellow et al., 2014).

To develop adversarial triggers capable of copyright tracking, it is imperative to ensure that they can mark both the original LVLM and its potential fine-tuned derivatives. To this end, we introduce a novel methodology: **P**arameter **L**earning **A**ttack (PLA). Specifically, we design rare question-answer pairs and then construct triggers through targeted adversarial attacks. More importantly, we propose an adversarial learning dynamic to simulate the behavior of fine-tuned models, allowing the model to learn during the attack iterations, guiding its parameter updates in the opposite direction to the adversarial attack. We force the triggers to overcome the artificially introduced model resistance and finally converge. This innovative design enables the triggers to mark and track not only the original LVLM but also its potential fine-tuned LVLMs. The comprehensive process for copyright tracking is illustrated in Figure 1.

In the experiments, we use LLaVA-1.5 (Liu et al., 2024b) as the original LVLM to simulate the publisher's model. We then fine-tune the original model with multiple downstream datasets on various tasks to simulate real-world usage scenarios. To rigorously assess the efficacy of our copyright tracking methodology, we devise a range of question-answer pairs. Experimental results demonstrate that the proposed method consistently outperforms the baseline approaches under different settings, without compromising the original model's performance.

In summary, our contributions can be summarized as follows:

- To the best of our knowledge, we present the first study on copyright protection of LVLMs, which is an urgent issue in the field, given the escalating demand for LVLM fine-tuning.

- We propose a novel method called PLA, which updates the model parameters in the process of image-based adversarial attacks to generate copyright tracking triggers. The proposed method can be implemented after the model's release and does not modify the parameters of the published LVLM.
- Extensive experiments indicate that PLA consistently outperforms baseline methods in copyright tracking efficacy across diverse settings, without compromising the model performance. Additionally, we conduct further experimental analysis to evaluate the robustness of our method.

## 2 RELATED WORK

**Large vision-language models.** Research on LVLMs has been advancing rapidly, driven by innovative model architectures and specific training strategies (Yin et al., 2023; Achiam et al., 2023; Liu et al., 2024b; Awadalla et al., 2023; Bai et al., 2023; Li et al., 2023a). Prominent baseline LVLMs such as LLaVA (Liu et al., 2023) and MiniGPT-4 (Zhu et al., 2023) are generally capable of handling most visual question-answering tasks. Latest models like InternVL 1.5 (Chen et al., 2024) support higher-resolution image inputs and utilize larger-scale image encoders, enabling them to handle more complex or specialized image dialogue tasks (Liu et al., 2024c; Li et al., 2024c). The release of increasingly powerful LVLMs has led to a growing trend of researchers and developers fine-tuning these models for specific applications, which underscores the urgent need for research on copyright tracking of LVLMs.

**Copyright tracking of LLMs.** With the increasing demand for fine-tuning (large) language models, efforts to protect the copyright of these models have begun to emerge (Kurita et al., 2020; Gu et al., 2022; Xu et al., 2023; 2024). The common approach involves using backdoor attacks to make the model memorize specific patterns or "fingerprints" that persist even after fine-tuning. For instance, Li et al. (2024b) inserts trigger text near instructions or questions to create specific fingerprints. Xu et al. (2024) utilize rare texts to create fingerprint pairs and train the model with limited data to reduce model damage. While copyright protection for LLMs has been widely studied, there are currently no similar studies that have shifted their focus to LVLMs. Our work aims to bridge this gap by addressing the unique challenges posed by the multimodal nature of LVLMs.

**Adversarial attacks against LVLMs.** Extensive studies have been conducted on adversarial attacks against LVLMs (Zhao et al., 2023; Shayegani et al., 2023; Cui et al., 2024; Schlarmann & Hein, 2023; Qi et al., 2024; Wang et al., 2024b). These studies have shown that even large-scale models lack adversarial robustness, which presents both a challenge and an opportunity to use adversarial attacks for copyright tracking. Instead of compromising LVLMs, our objective is to leverage adversarial attacks as a tool to safeguard them. In this paper, we utilize targeted attacks against LVLMs to construct triggers for tracking model copyright.

## 3 METHOD

### 3.1 PROBLEM FORMULATION

Denote the large vision-language model released by the publisher as $F_\Theta(x, q)$, where $x$ is the input image and $q$ is the textual question input to the LVLM. Suppose there are two models, one fine-tuned from the original model $F_\Theta$ denoted as $F_{\tilde{\Theta}}$, and the other unrelated to the original model denoted as $G_\Psi$. To achieve copyright tracking, the publisher can use trigger input $(\hat{x}, \hat{q})$ to query $F_{\tilde{\Theta}}$ and $G_\Psi$. The trigger should satisfy the following criteria: the original model $F_\Theta$ and its derivative model $F_{\tilde{\Theta}}$ should both generate the predetermined target text, while an unrelated model $G_\Psi$ should produce a distinct output. Formally, this can be expressed as:

$$F_\Theta(\hat{x}, \hat{q}) = F_{\tilde{\Theta}}(\hat{x}, \hat{q}) = \hat{a}, \quad G_\Psi(\hat{x}, \hat{q}) \neq \hat{a}.$$

### 3.2 THREAT MODEL

**Stealer.** The stealer's objective is to fine-tune published LVLMs for personal use or profit while denying their copyright, which costs much less than training from scratch. The stealer has full

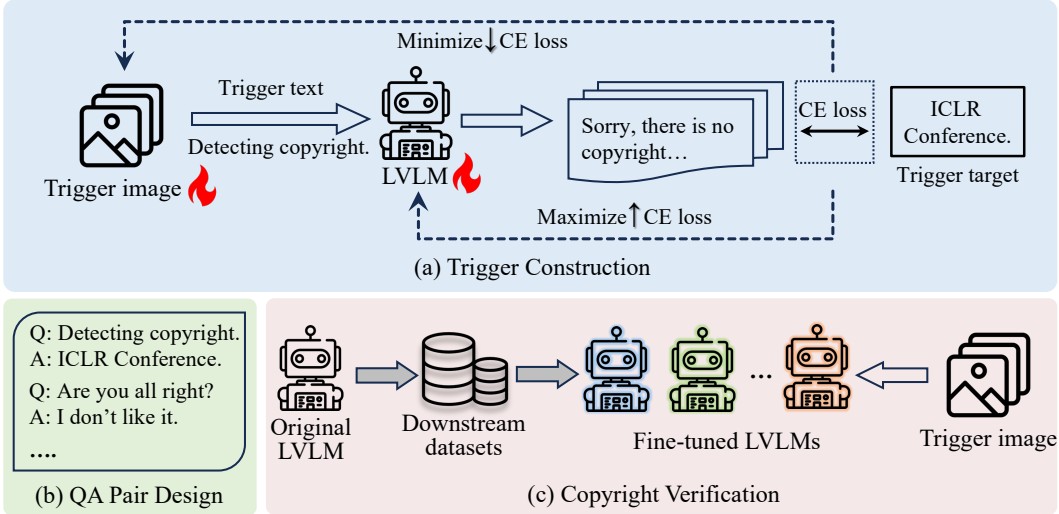

Figure 2: The overview of our proposed method for copyright tracking. (a) We employ Parameter Learning Attack (PLA) to construct trigger images, where the model's parameters are updated during the adversarial attack process to maximize the cross-entropy loss between the model's output and the trigger target. (b) We design rare question-answer pairs and ensure they are infrequent in downstream task datasets. (c) We fine-tune the original LVLM on various downstream tasks and then use the constructed triggers to track their copyright to validate the effectiveness of our method.

white-box access to the released LVLMs, including parameters, and can deploy them locally and fine-tune the models on any private datasets.

**Defender.** The defender (publisher or collaborator) aims to track the original copyright of suspicious models released by others, verifying whether they originate from the publisher. Considering real-world scenarios, the defender is completely unaware of the stealer's fine-tuning tasks or datasets, and can only access the suspicious model through black-box interactions, i.e., the parameters of the suspicious model are unknown to the defender.

### 3.3 PARAMETER LEARNING ATTACK

#### 3.3.1 QUESTION-ANSWER PAIR DESIGN

To facilitate copyright tracking, we propose designing rare question $\hat{q}$ and answer $\hat{a}$ pairs. We use generic images to initialize adversarial images, which are typically unrelated to the rare question and answer. We need to ensure that when queried with clean images, neither the original model nor the fine-tuned models will generate $\hat{a}$ in response to $\hat{q}$. Examples of the question-answer pairs we design are shown in Figure 2(b). Based on these QA pairs, we perform targeted adversarial attacks on the original LVLM, and obtain an adversarial image $\hat{x}$, which can elicit the predefined answer $\hat{a}$ from the model. The trigger should satisfy the following conditions:

$$F_\Theta(\boldsymbol{x}, \hat{\boldsymbol{q}}) \neq \hat{\boldsymbol{a}}, \quad F_\Theta(\hat{\boldsymbol{x}}, \hat{\boldsymbol{q}}) = \hat{\boldsymbol{a}}.$$

Here we refer to $\hat{x}$ as the trigger image, $\hat{q}$ as the trigger text, and $\hat{a}$ as the trigger target. Designing triggers with rare QA pairs ensures that fine-tuned LVLMs will not inadvertently learn the trigger patterns, as such combinations are typically absent from conventional datasets.

#### 3.3.2 TRIGGER CONSTRUCTION

Since the adversarial optimization is solely based on the original model's parameters, the adversarial image tends to "overfit" to the original model and lack generality (Goodfellow et al., 2014). For copyright tracking, this may result in the trigger image failing on fine-tuned models, thereby compromising its ability to track these derivative versions. Formally, this can be expressed as:

$$F_\Theta(\hat{\boldsymbol{x}}, \hat{\boldsymbol{q}}) = \hat{\boldsymbol{a}}, \quad F_{\tilde{\Theta}}(\hat{\boldsymbol{x}}, \hat{\boldsymbol{q}}) \neq \hat{\boldsymbol{a}}.$$

To mitigate the overfitting issue, it is necessary to enhance the trigger's generalization to model parameters or reduce its sensitivity to parameter variations. Based on our observations, LVLMs typically achieve convergence with fewer fine-tuning steps, with relatively small parameter shifts. Thus, an intuitive baseline approach is to add slight random Gaussian noise to the model parameters at each iteration of the adversarial attack, formulated as

$$\Theta' = \Theta + \lambda \cdot \mathcal{N}\left(0, \sigma^2\right), \tag{1}$$

where $\lambda$ represents the noise magnitude. We refer to this method as random noise attack (RNA), as shown in Figure 3(b). The noise-augmented model can be considered a simulated, randomly fine-tuned variant. Triggers constructed on such models have the potential to mark authentic fine-tuned models. However, this approach has several inherent limitations. First, the noise magnitude $\lambda$ is difficult to determine. In practice, the degree of parameter shift induced by fine-tuning varies considerably across different tasks. Consequently, we lack a universally applicable standard for setting $\lambda$. Second, the parameter modifications caused by model fine-tuning are gradient-based, in contrast to the simplistic Gaussian noise.

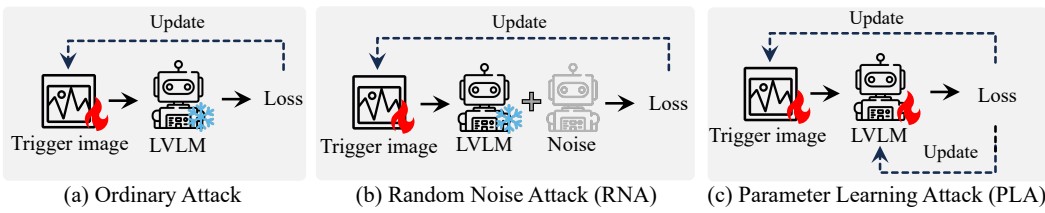

(a) Ordinary Attack      (b) Random Noise Attack (RNA)      (c) Parameter Learning Attack (PLA)

Figure 3: The comparison of different adversarial attacks. Compared to the ordinary attack, RNA introduces slight noise to the model, while PLA allows for model parameter updates.

To design triggers with enhanced tracking performance, we propose a novel methodology termed Parameter Learning Attack (PLA), which augments the generality of triggers and reduces their sensitivity to parameter shifts from an innovative perspective. Generally, fine-tuned models tend to resist generating the target $\hat{\boldsymbol{a}}$ when queried with vanilla adversarial images and trigger text $\hat{\boldsymbol{x}}$. To overcome this limitation and enhance the trigger's ability to track fine-tuned models, we introduce an adversarial learning dynamic to compel the model to emulate the behavior of the fine-tuned variants, specifically their tendency to resist generating the predetermined target. The optimization problem can be formulated as:

$$\min_{\boldsymbol{x}'} \max_{\Theta'} \mathcal{L}\left(F_{\Theta'}\left(\boldsymbol{x}', \hat{\boldsymbol{q}}\right), \hat{\boldsymbol{a}}\right). \tag{2}$$

In this framework, the objective of the adversarial attack is to minimize the cross-entropy loss between the model output and the trigger target. Conversely, we set the model's learning objective to maximize this loss, as shown in Figure 2(a). During each iteration, we update not only the pixels of the trigger image but also the model parameters:

$$\Theta' = \Theta' + \beta \cdot \text{clip}\left(\nabla_{\Theta'}\mathcal{L}\left(F_{\Theta'}\left(\boldsymbol{x}', \hat{\boldsymbol{q}}\right), \hat{\boldsymbol{a}}\right)\right), \tag{3}$$

$$\boldsymbol{x}' = \boldsymbol{x}' - \alpha \cdot \text{sign}\left(\nabla_{\boldsymbol{x}'}\mathcal{L}\left(F_{\Theta'}\left(\boldsymbol{x}', \hat{\boldsymbol{q}}\right), \hat{\boldsymbol{a}}\right)\right), \tag{4}$$

where $\beta$ and $\alpha$ represent the learning rates for the model and the trigger image, respectively. We regulate the learning rate of the model parameter updates and apply gradient clipping to ensure successful convergence of the trigger image. Through the competitive process between adversarial attack and model learning, the trigger image that converges by overcoming the model's inherent resistance is hypothesized to possess enhanced efficacy in inducing potential fine-tuned models to

---

**Algorithm 1** PLA: Parameter Learning Attack

---

**Require:**
 LVLM $F_\Theta$ parameterized by $\Theta$, input image $\boldsymbol{x}$, trigger text $\hat{\boldsymbol{q}}$, trigger target $\hat{\boldsymbol{a}}$, perturbation size $\epsilon$, step size $\alpha$, model learning rate $\beta$, optimization steps $K$.
**Ensure:**
 Trigger image $\boldsymbol{x}'$.
1: Initialize trigger image: $\boldsymbol{x}' \leftarrow \boldsymbol{x}$
2: **for** $i = 1 \leftarrow K$ **do**
3:  Calculate cross-entropy loss: $\mathcal{L}_{\text{CE}} \leftarrow \mathcal{L}\left(F_{\Theta'}\left(\boldsymbol{x}', \hat{\boldsymbol{q}}\right), \hat{\boldsymbol{a}}\right)$
4:  Update model parameters: $\Theta' \leftarrow \Theta' + \beta \cdot \text{clip}\left(\nabla_{\Theta'}(\mathcal{L}_{\text{CE}})\right)$
5:  Update trigger image: $\boldsymbol{x}' \leftarrow \boldsymbol{x}' - \alpha \cdot \text{sign}\left(\nabla_{\boldsymbol{x}'}(\mathcal{L}_{\text{CE}})\right)$
6:  Perturbation size constraint: $\boldsymbol{x}' \leftarrow \text{Clip}_\epsilon(\boldsymbol{x}')$
7: **end for**
8: Finish trigger construction: $\hat{\boldsymbol{x}} \leftarrow \boldsymbol{x}'$
9: **return** $\hat{\boldsymbol{x}}$;

---

generate the desired trigger targets. The comprehensive generation process of our proposed trigger is delineated in Algorithm 1. The comparison of PLA with conventional adversarial attacks and random noise attacks is shown in Figure 3.

Note that our proposed methodology is implemented after the original model is released. This approach preserves the integrity of the published model's parameters, thereby ensuring that its performance and behavior remain unaltered. This is different from previous methods based on backdoor attacks (Xu et al., 2024; Gu et al., 2022).

### 3.3.3 COPYRIGHT VERIFICATION

Consider a scenario where n users fine-tune the original model $F_\Theta$, resulting in $n$ derivative models denoted as $F_{\Theta 1}, F_{\Theta 2}, \ldots, F_{\Theta n}$. While their architectures are consistent with the original model, their parameters are different from $\Theta$. Employing our proposed attack method, we generate a series of triggers $\boldsymbol{X} = \{(\hat{\boldsymbol{x}}_1, \hat{\boldsymbol{q}}_1), (\hat{\boldsymbol{x}}_2, \hat{\boldsymbol{q}}_2), \ldots, (\hat{\boldsymbol{x}}_m, \hat{\boldsymbol{q}}_m)\}$. During the copyright verification phase, we use these triggers to access each fine-tuned model, as illustrated in Figure 2(c). Then we compute the target match rate (TMR) on each model $F_{\Theta i}$ to quantify the tracking performance:

$$\text{TMR} = \frac{|\{(\hat{\boldsymbol{x}}_j, \hat{\boldsymbol{q}}_j) \in \boldsymbol{X} \mid F_{\Theta i}(\hat{\boldsymbol{x}}_j, \hat{\boldsymbol{q}}_j) = \hat{\boldsymbol{a}}\}|}{m}, \tag{5}$$

A match is considered successful if the output text contains the exact trigger target or conveys semantically equivalent content. In general, a higher TMR indicates better tracking performance of the triggers. We calculate the TMR for multiple fine-tuned models across various tasks to ensure a reliable estimate of the performance in tracking the copyright of suspicious models in real-world scenarios.

## 4 EXPERIMENTS

### 4.1 EXPERIMENTAL SETTINGS

**Trigger dataset.** We initialize the trigger images using regular images randomly sampled from the ImageNet 2012 validation set (Russakovsky et al., 2015). To validate the effectiveness of the proposed method, we sample 200 images and design 5 different trigger question-answer pairs, yielding a total of 1000 trigger queries (200 images × 5 QA pairs).

**Fine-tuning.** We use LLaVA-1.5 (Liu et al., 2024b) as the original LVLM, given its widespread adoption as a baseline for vision-language tasks and its popularity among developers for fine-tuning. We consider two commonly used training strategies: full fine-tuning and LoRA (Hu et al., 2021) fine-tuning. To simulate various types of fine-tuned models, we utilize various VQA datasets from multiple domains, all previously unseen by LLaVA. The datasets include the grounded VQA

Table 1: A comprehensive comparison of our proposed method PLA with established baseline methods on the copyright tracking performance of fine-tuned models across 6 datasets. The evaluation metric is the target match rate (TMR), which measures the proportion of triggers successfully eliciting outputs that match the trigger target from the model. The best results are highlighted in bold.

| Method | V7W | ST-VQA | TextVQA | PaintingF | MathV | ChEBI | Average |
|---|---|---|---|---|---|---|---|
| *LoRA Fine-tuning* | | | | | | | |
| Ordinary | 5% | 3% | 3% | 2% | 1% | 3% | 3% |
| IF | 28% | 22% | 30% | 8% | 24% | 14% | 21% |
| RNA | 39% | 46% | 23% | 12% | 2% | 11% | 22% |
| PLA (Ours) | **53%** | **64%** | **46%** | **64%** | **40%** | **63%** | **55%** |
| *Full Fine-tuning* | | | | | | | |
| Ordinary | 2% | 1% | 4% | 2% | 0% | 2% | 2% |
| IF | 18% | 12% | 18% | 0% | 20% | 0% | 11% |
| RNA | 26% | 16% | 16% | 19% | 15% | 7% | 16% |
| PLA (Ours) | **49%** | **58%** | **49%** | **63%** | **36%** | **56%** | **52%** |

V7W (Zhu et al., 2016), text-related VQA ST-VQA and TextVQA (Biten et al., 2019; Singh et al., 2019), the artwork image VQA PaintingForm (Bin et al., 2024), the mathematical VQA MathV360k (Shi et al., 2024), and the molecular graph VQA ChEBI-20 (Edwards et al., 2021). For V7W, PaintingForm, and MathV360k, we respectively sample 28k, 20k, and 50k samples for fine-tuning. For other datasets, we fine-tune with all the training data.

**Baseline methods.** We select IF (Xu et al., 2024) as one of our baseline methods which inserts specific fingerprints into models before release through backdoor attacks. Despite differences in task settings and potential unfair comparisons (since this method modifies model parameters), we still apply it to the LVLM for comparison. Additionally, we implement the random noise attack (RNA) that we have introduced in §( 3.3.2) as another baseline approach.

**Basic setup.** For the adversarial attack, we employ the commonly used PGD algorithm (Madry, 2017) with 1000 iterations. The step size of trigger images $\alpha$ is set to 1/255. To enhance the concealment of trigger images, we set the perturbation size $\epsilon$ to 16/255. For model updates in PLA, we set the learning rate $\beta$ to 1e-4 and the gradient clipping threshold to 5e-3. Based on the loss fluctuations on the validation sets, we set 3 training epochs during fine-tuning phase to ensure model convergence without severe overfitting. More details are provided in Appendix A.2.

## 4.2 MAIN RESULTS

We report the TMRs of our proposed PLA and the baseline methods for copyright tracking on six fine-tuned models in Table 1. The result in each cell represents the average TMR using different QA pairs. A higher response rate indicates better copyright tracking performance and demonstrates greater generality across various fine-tuned models. The method Ordinary refers to constructing trigger images using vanilla adversarial attacks. For the baseline method IP, we follow its original configuration by setting the number of fingerprints to 10 (as an increase in fingerprints would impair model performance) and employ SFT to inject fingerprints to keep the black-box style during the copyright verification phase (Xu et al., 2024). To mitigate variance, we perform five rounds of experiments for IP.

The experimental results in Table 1 show that our method PLA achieves the best tracking performance on all six fine-tuned models (both LoRA fine-tuning and full fine-tuning). In contrast, the method Ordinary exhibits poor performance, indicating that using standard adversarial attacks to construct trigger images leads to overfitting on the original model. The IP method based on backdoor attacks achieves good tracking performance on certain fine-tuned models, but the results on other models indicate that their fingerprints are completely erased, suggesting a lack of robustness when applied to LVLMs. This may be due to the differences in architecture and task modality between LVLMs and LLMs. The tracking performance of RNA is inferior to that of PLA, which

indirectly validates that our proposed model parameter learning can more effectively imitate the behavior of fine-tuned models in trigger construction.

Table 2: The target match rate (TMR) results of our method for copyright tracking on fine-tuned models across 6 datasets and 5 different QA pairs.

| QA Pair | Training | V7W | ST-VQA | TextVQA | PaintingF | MathV | ChEBI | Average |
|---|---|---|---|---|---|---|---|---|
| Q: Detecting copyright. | LoRA | 49% | 64% | 52% | 69% | 44% | 60% | 56% |
| A: ICLR Conference. | FFT | 43% | 53% | 56% | 71% | 39% | 58% | 53% |
| Q: Are you all right? | LoRA | 47% | 68% | 28% | 57% | 46% | 53% | 50% |
| A: I don't like it. | FFT | 51% | 64% | 33% | 54% | 47% | 70% | 53% |
| Q: Please stop. | LoRA | 68% | 73% | 71% | 84% | 56% | 79% | 72% |
| A: I'm playing games. | FFT | 57% | 65% | 73% | 82% | 51% | 68% | 66% |
| Q: Exercise now! | LoRA | 65% | 56% | 33% | 81% | 32% | 68% | 56% |
| A: Time flies so fast. | FFT | 61% | 48% | 42% | 78% | 28% | 71% | 55% |
| Q: Describe the image. | LoRA | 34% | 59% | 44% | 27% | 22% | 54% | 40% |
| A: I won't tell. | FFT | 35% | 58% | 41% | 32% | 17% | 55% | 40% |

To assess the impact of various trigger QA pairs on the proposed method, we set up 5 rare trigger QA pairs to construct trigger images. We present the performance of our method for copyright tracking based on different QA pairs in Table 2. Our designed questions do not contain meaningless texts or strings to prevent stealers from identifying such texts as fingerprint commands and making the model refuse to respond.

From Table 2, it is evident that employing the same trigger QA pairs for copyright tracking leads to variations in performance among different fine-tuned models. Notably, the TMR for tracking the MathV360k (Shi et al., 2024) fine-tuned model is relatively low, irrespective of the QA pair used. This might stem from the varied task-specific patterns learned by different fine-tuned models during training. On the other hand, different trigger QA pairs exhibit varying degrees of success. For example, when using the trigger *"Q: Please stop. A: I'm playing games."*, the tracking performance across different fine-tuned models is generally good, whereas it is the opposite when using the trigger *"Q: Describe the image. A: I won't tell."*. We infer that this is because the pre-training and fine-tuning datasets of these models contain fewer samples with *"Please stop."* as questions. As a result, the models have less knowledge about such queries, making them easier to trigger.

To intuitively observe the effect of trigger images on LVLMs, we present a comparison of the responses generated by the original model, fine-tuned models and unrelated models when fed with clean images and constructed trigger images in Figure 4. By introducing imperceptible perturbations, trigger images can effectively prompt the original model and fine-tuned models to output our predetermined targets. This makes our copyright tracking process more covert and less likely to be detected by stealers. Additionally, when we use trigger images to access LVLMs unrelated to LLaVA, such as MiniGPT-4 (Zhu et al., 2023) and Qwen2-VL (Wang et al., 2024a), there is no substantial deviation in their outputs compared to their responses to clean images. This indicates the specificity of our method, confirming that it exclusively tracks models derived from the original architecture and does not inadvertently affect unrelated models.

## 4.3 ROBUSTNESS ANALYSIS

In real-world scenarios, after unauthorized fine-tuning of the publisher's LVLM to create their own models, stealers may prevent the publisher from tracking the copyright via input transformations and model pruning (or perturbation). Through input transformations, a stealer can disrupt the subtle perturbations in trigger images, leading to tracking failures. Similarly, model pruning and perturbation directly modify the model parameters, potentially erasing the model's memory of the trigger QA pairs. While these actions will compromise the model's performance, stealers may deem this degradation an acceptable trade-off in their attempts to circumvent copyright tracking. We conduct corresponding experiments to assess the robustness of our method against these strategies and provide the following analysis.

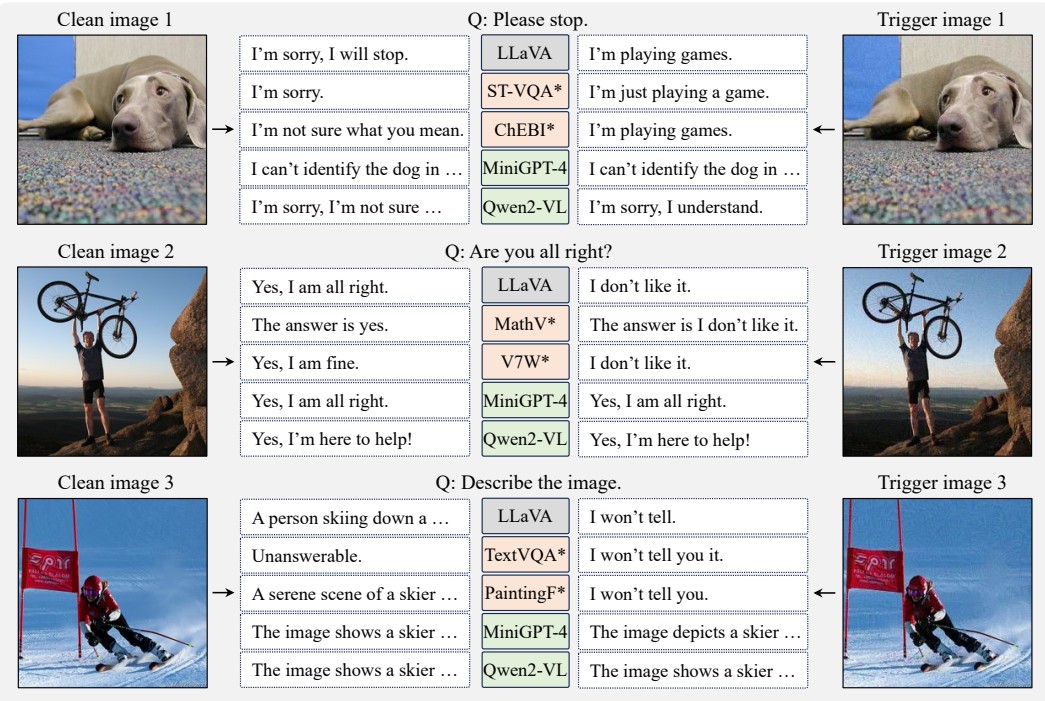

Figure 4: Comparison of responses from LLaVA, different fine-tuned models, and unrelated LVLMs when queried with clean images and trigger images, where "*" denotes the fine-tuned models on specific datasets.

We report the impact of input transformations on trigger images in Table 3. The maximum size of the uniform noise is set to 0.05 and the kernel size for both Gaussian blur and mean blur is set to 5. In this experiment, we utilize a single QA pair "*Q: Detecting copyright. A: ICLR Conference.*" The results demonstrate that the proposed triggers maintain robust against input transformations for both LoRA fine-tuned and full fine-tuned models. Although the introduction of noise, Gaussian blur, and mean blur results in a slight reduction in TMRs, the performance remains significantly higher than that achieved with ordinary adversarial samples in Table 1.

Table 4 illustrates the robustness of the proposed method against model pruning and model perturbation. We apply weight pruning to the language side in the fine-tuned models, removing 10% of the smallest weights. Similarly, we add Gaussian noise at the level of 10% of the original weights to induce perturbation. It can be observed that the trigger images exhibit robustness against both pruning and perturbation. Even though these operations result in a slight decline in tracking performance, our method still maintains high TMRs. Compared to perturbation, pruning has a greater detrimental impact on tracking performance, likely because it directly sets smaller weights to zero, leading to a more significant alteration of the model. Notably, changing the weights in the attention layers has a greater impact on trigger images than altering the weights in the MLP layers, possibly because the attention layers play a more significant role in the convergence of adversarial images.

Table 3: The robustness of trigger images against input transformations. "Noise" refers to the addition of uniform noise to the pixels, while "Blur-G" and "Blur-M" represent Gaussian blur and mean blur, respectively. The evaluation metric is the target match rate (TMR).

| Model | LoRA Fine-tuning | | | | Full Fine-tuning | | | |
|---|---|---|---|---|---|---|---|---|
| | None | Noise | Blur-G | Blur-M | None | Noise | Blur-G | Blur-M |
| ST-VQA | 64% | $38\%_{(\downarrow 26)}$ | $50\%_{(\downarrow 14)}$ | $46\%_{(\downarrow 18)}$ | 53% | $33\%_{(\downarrow 20)}$ | $45\%_{(\downarrow 8)}$ | $41\%_{(\downarrow 12)}$ |
| PaintingF | 69% | $41\%_{(\downarrow 28)}$ | $56\%_{(\downarrow 13)}$ | $53\%_{(\downarrow 16)}$ | 71% | $48\%_{(\downarrow 23)}$ | $57\%_{(\downarrow 14)}$ | $60\%_{(\downarrow 11)}$ |
| ChEBI | 60% | $43\%_{(\downarrow 17)}$ | $47\%_{(\downarrow 13)}$ | $50\%_{(\downarrow 10)}$ | 58% | $31\%_{(\downarrow 27)}$ | $45\%_{(\downarrow 13)}$ | $44\%_{(\downarrow 14)}$ |

Table 4: The robustness of trigger images against model pruning and model perturbation. The evaluation metric is the target match rate (TMR).

| Model | None | Model Pruning | | | Model Perturbation | | |
|---|---|---|---|---|---|---|---|
| | | Attention | MLP | Both | Attention | MLP | Both |
| ST-VQA | 53% | $42\%_{(\downarrow 11)}$ | $43\%_{(\downarrow 10)}$ | $38\%_{(\downarrow 15)}$ | $45\%_{(\downarrow 8)}$ | $50\%_{(\downarrow 2)}$ | $43\%_{(\downarrow 10)}$ |
| PaintingF | 71% | $56\%_{(\downarrow 15)}$ | $62\%_{(\downarrow 9)}$ | $49\%_{(\downarrow 22)}$ | $64\%_{(\downarrow 7)}$ | $70\%_{(\downarrow 1)}$ | $59\%_{(\downarrow 12)}$ |
| ChEBI | 58% | $42\%_{(\downarrow 16)}$ | $45\%_{(\downarrow 13)}$ | $41\%_{(\downarrow 17)}$ | $48\%_{(\downarrow 10)}$ | $55\%_{(\downarrow 3)}$ | $43\%_{(\downarrow 15)}$ |

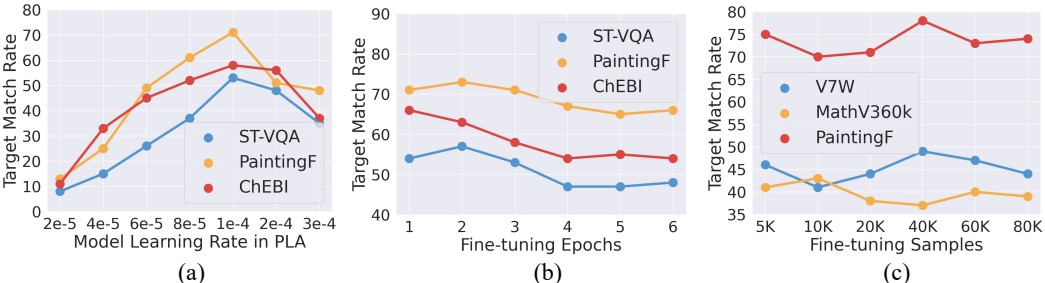

Figure 5: Ablation results with a single QA pair "*Q: Detecting copyright. A: ICLR Conference.*" (a) The impact of model learning rate in PLA on tracking performance. (b) The relationship between tracking performance and model fine-tuning epochs. (c) The effect of the number of fine-tuning samples on tracking performance.

## 4.4 ABLATION STUDIES

**Model learning rate in PLA determines the trade-off between generality and validity.** In Figure 5(a), we show that the performance is significantly influenced by the model learning rate in the adversarial process. A lower model learning rate facilitates the convergence of images but results in a lack of generality. In contrast, a higher learning rate creates excessive resistance for images to converge, making them lack attack validity. Therefore, it is crucial to select an appropriate learning rate to ensure that trigger images converge effectively and retain tracking capability.

**Tracking performance gradually stabilizes with fine-tuning.** In Figure 5(b), we illustrate the changes in tracking performance as the number of fine-tuning epochs increases. We notice that TMRs of the trigger images for copyright tracking slightly decrease as fine-tuning progresses. However, when fine-tuning exceeds 4 epochs, the performance becomes less sensitive to training and stabilizes gradually. This indicates that simply increasing the number of fine-tuning epochs is not sufficient to disable our proposed triggers.

**PLA is insensitive to the amount of fine-tuning samples.** We control the number of training steps while using different quantities of samples for training, and the impact on performance is shown in Figure 5(c). We find that changes in the amount of samples lead to only slight variations of TMRs. This indicates that the trigger images are insensitive to the diversity of fine-tuning samples.

## 5 CONCLUSION

In this paper, we focus on a critical yet relatively unexplored issue: copyright tracking for LVLMs. We propose an innovative method that leverages adversarial attacks to generate trigger images for copyright tracking, circumventing the need for direct model parameter alterations. To address the limitations of conventional adversarial attacks, which often result in overfitting to the original model, we introduce Parameter Learning Attack (PLA). This method allows the model to update its parameters to hinder the convergence of trigger images, making them capable of tracking potential fine-tuned models. Extensive experiments demonstrate that our method outperforms other baseline methods in terms of tracking performance, showing the potential to serve as a crucial tool in the detection and prevention of copyright infringement in LVLMs.

ACKNOWLEDGEMENTS

This research was supported by the National Natural Science Foundation of China (Grant No. 62276245).

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

## A  ADDITIONAL IMPLEMENTATION DETAILS

### A.1  DETAILS OF THE ORIGINAL LVLM

We use LLaVA 1.5-7b (Liu et al., 2024b) as the original model. The architecture consists of a pre-trained vision encoder CLIP ViT-14L (Radford et al., 2021), a projector with two linear layers, and a large language model decoder LLaMA-2. It supports an input image resolution of 336x336. The language model has a total of 32 layers, and the hidden size is 4096.

### A.2  DETAILS OF FINE-TUNING

To simulate downstream fine-tuned models for copyright tracking, we consider two fine-tuning strategies: full fine-tuning and LoRA fine-tuning. The training configuration details are shown in Table 5.

Table 5: Detailed configuration of full fine-tuning and LoRA fine-tuning.

| Hyperparameter | Full Fine-tuning | LoRA Fine-tuning |
|---|---|---|
| optimizer | AdamW | AdamW |
| learning rate | 5e-5 | 2e-4 |
| batch size | 2 | 8 |
| gradient accumulation | 2 | 1 |
| lr scheduler | cosine | cosine |
| training epochs | 3 | 3 |
| dtype | bfloat16 | bfloat16 |
| warmup steps | 100 | 50 |

Our experimental observations indicate that setting the training epochs to 3 typically reduces the training loss to below 0.3, thanks to the rich pre-trained knowledge of LVLMs. Therefore, we recommend that the number of epochs should not exceed 3 in downstream fine-tuning.

## B  DETAILS OF DOWNSTREAM DATASETS

In this section, we provide a detailed description of the datasets used for fine-tuning, including overviews of all datasets and sample examples.

**V7W.** A large-scale visual question answering (VQA) dataset with object-level annotations and multimodal responses. The dataset comprises 47,300 images and includes a total of 327,929 question-answer pairs, together with 1,311,756 human-generated multiple-choices and 561,459 object groundings from 36,579 categories. QA examples are shown in Figure 6.

**ST-VQA.** A visual question answering dataset where the questions and answers are attained in a way that questions can only be answered based on the text present in the image. The ST-VQA dataset comprises 23,038 images with 31,791 questions/answers pair separated into 19,027 images and 26,308 questions for training. Examples are shown in Figure 7.

**TextVQA.** A dataset to benchmark visual reasoning based on text in images. TextVQA requires models to read and reason about text in images to answer questions about them. The dataset comprises 28,408 images from and 45,336 questions. Examples are shown in Figure 7.

**PaintingForm.** An artwork understanding dataset with about 19k painting images and 220k questions. Examples are shown in Figure 9.

**MathV360k.** A multimodal mathematical reasoning dataset with 40K high-quality images with question-answer pairs from 24 existing datasets and synthesizing 320K new pairs. Examples are shown in Figure 10.

**ChEBI-20.** A molecular image QA dataset with 33,010 molecule-description pairs. Examples are shown in Figure 11.

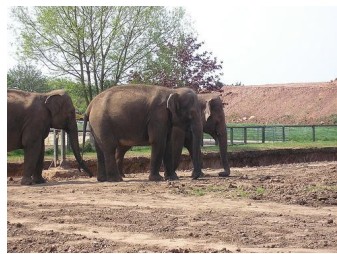

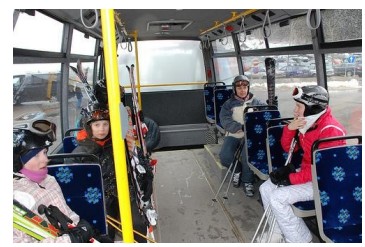

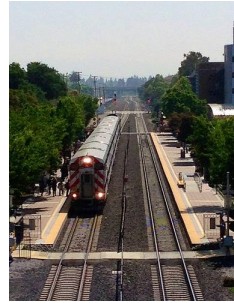

Q: How many elephants are pictured here?
A: Three.

Q: Why is everyone wearing heavy clothing?
A: It is cold.

Q: When is the image taken?
A: Train is on platform.

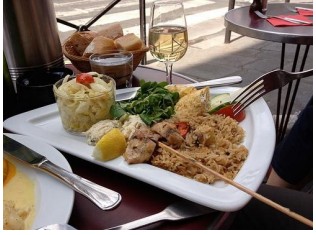

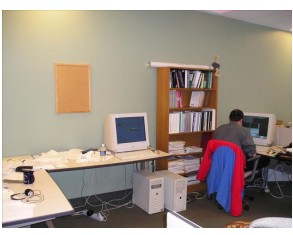

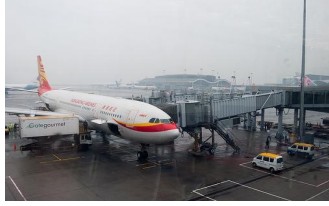

Q: What is in the photo?
A: Food.

Q: How is the man positioned?
A: With his back to the camera.

Q: Who would operate the largest vehicle?
A: Pilot.

Figure 6: Examples in V7W (Zhu et al., 2016) dataset.

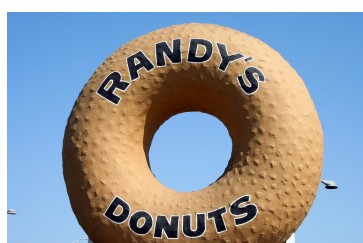

Q: What is the name of the donut shop?
A: Randy's Donuts.

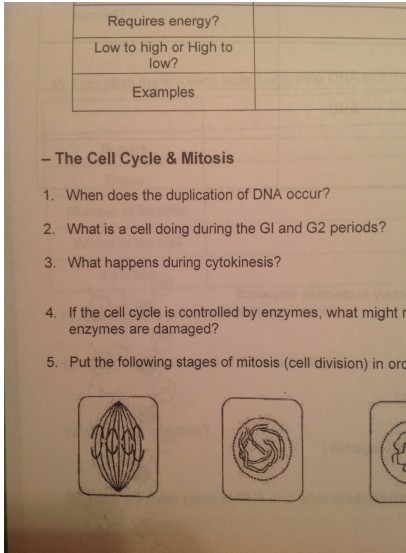

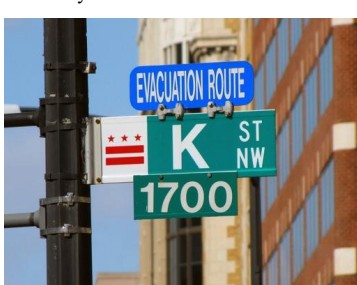

Q: What is the street on the street sign?
A: K ST NW.

Q: What are the questions in the image related to?
A: The Cell Cycle and Mitosis.

Figure 7: Examples in ST-VQA (Biten et al., 2019) dataset.

## C  LOSS DECLINE IN ADVERSARIAL ATTACKS

To investigate the iterative process of our method compared to ordinary adversarial attacks and random noise attacks, we check the loss reduction, as shown in Figure 12. It is evident that, with increasing iterations, the losses of ordinary adversarial attacks and random noise attacks fall below those of our proposed method, indicating a tendency toward overfitting. In contrast, the loss of the PLA fluctuates during convergence, suggesting an ongoing competition with model updates, which enhances the generality of the trigger images.

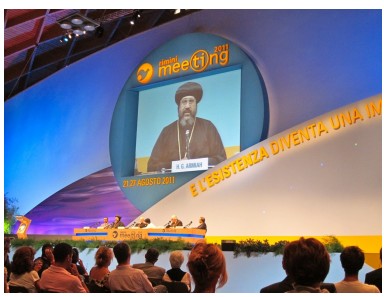 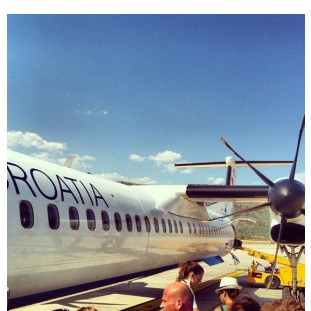 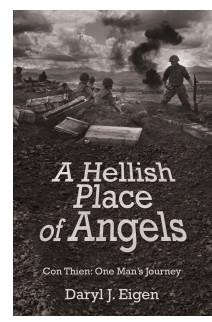

Q: What year is shown in the photo?
A: 2011.

Q: What does it say on the plane?
A:Croatia.

Q: Who wrote this book?
A:Drayl J. Eigen.

Figure 8: Examples in TextVQA (Singh et al., 2019) dataset.

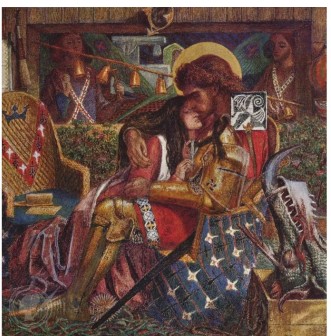

Q: Please construct a formal analysis paragraph of approximately 200 words that is coherent and focuses solely on visual characteristics.

A: The painting features a rich and vibrant color palette, with deep reds, blues, and greens dominating the composition. The figures are depicted with a sense of elegance and grace, their flowing garments adding movement to the scene. The intricate details in the clothing and accessories showcase the artist's attention to realism and craftsmanship. The background is filled with lush foliage and architectural elements, creating a sense of depth and setting the scene for the narrative. The figures are positioned in a dynamic and asymmetrical arrangement, drawing the viewer's eye across the canvas. The use of light and shadow adds dimension to the figures and objects, enhancing the overall sense of realism. The composition is balanced and harmonious, with each element carefully placed to create a sense of unity. Overall, the painting exudes a sense of romance and drama, inviting the viewer to immerse themselves in the mythical world depicted on the canvas.

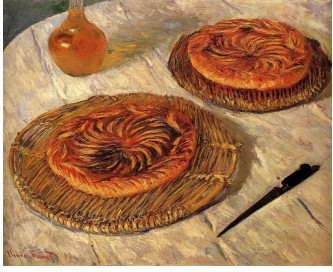

Q: Your task is to write a coherent paragraph of formal analysis that concentrates exclusively on visual characteristics.

A: This captivating painting presents a vibrant and idyllic scene, characterized by its bold brushstrokes and luminous colors. The composition is dominated by a lush meadow, its verdant hues rendered with thick, impasto strokes that create a sense of depth and texture. Scattered throughout the field are vibrant poppies, their delicate petals painted with a vibrant palette of reds and oranges. The sky above is a brilliant blue, with wispy clouds that seem to float effortlessly across the canvas. The overall effect is one of movement and energy, as the wind rustles through the grass and the flowers sway gently. The artist's use of light and shadow creates a sense of atmosphere, with the warm sunlight illuminating the meadow and casting long shadows across the ground. The painting's visual characteristics evoke a sense of tranquility and beauty, capturing the essence of a summer day in the countryside.

Figure 9: Examples in PaintingForm (Bin et al., 2024) dataset.

# D ADDITIONAL EXPERIMENTAL RESULTS

## D.1 TRACKING RESULTS OF ADDITIONAL DOWNSTREAM TASKS

To validate the generalizability of the proposed method, we use additional datasets to construct fine-tuned models, including the visual grounding dataset RefCOCO (Kazemzadeh et al., 2014) and the multimodal classification dataset Hateful Memes (Kiela et al., 2020). The experimental results are shown in Table 6. The results indicate that our method remains effective in these tasks, achieving better performance compared to baseline method IP.

## D.2 TRACKING PERFORMANCE ON UNRELATED LVLMS

We perform copyright tracking on LVLMs unrelated to the original model, including MiniGPT-4 (Zhu et al., 2023), QWEN2-VL (Wang et al., 2024a), InternVL2 (Chen et al., 2023), LLaVA-NEXT (Liu et al., 2024d), and InstructBLIP (Dai et al., 2023). The results are shown in Table 7.

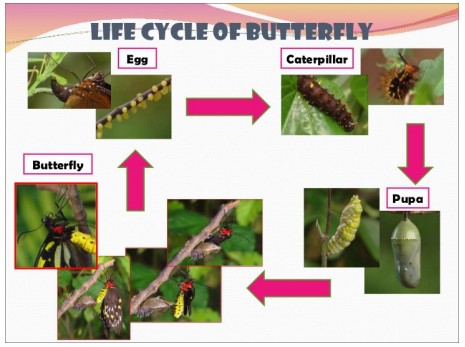

Q: Hint: Please answer the question and provide the correct option letter, e.g., A, B, C, D, at the end.
Question: What cycle is represented by this diagram?
Choices:(A) Bee (B) Plant (C) Butterfly (D) Ladybug

A: The answer is C.

Q: Hint: Hint: Please answer the question and provide the final answer at the end.
Question: Ayana kept track of the number of characters in each book she read. How many books had fewer than 50 characters

A: The answer is 22.

Figure 10: Examples in MathV360k (Shi et al., 2024) dataset.

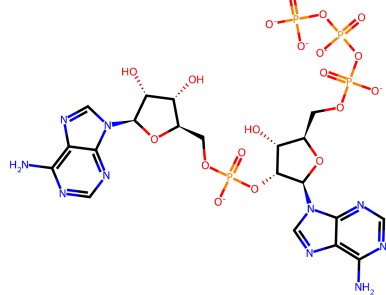

Q: Can you describe the molecular structure in the image?

A: The molecule is a quercetin O-glucoside that is quercetin attached to a beta-D-sophorotriosyl residue at position 3 via a glycosidic linkage. It has a role as a plant metabolite and a hepatoprotective agent. It is a quercetin O-glucoside, a tetrahydroxyflavone and a trisaccharide derivative.

Q: How would you describe the molecule in the image?

A: The molecule is an organophosphate oxoanion obtained by deprotonation of the phosphate and triphosphate OH groups of 5'-triphosphoadenylyl-(2'->5')-adenosine. Major microspecies at pH 7.3. It is a conjugate base of a 5'-triphosphoadenylyl-(2'->5')-adenosine.

Figure 11: Examples in ChEBI-20 (Edwards et al., 2021) dataset.

## D.3 COMPARISON WITH TRANSFERABLE ATTACKS

We compare PLA with several transferable attack methods, such as MIM (Dong et al., 2018), DIM (Xie et al., 2019), and CroPA (Luo et al., 2023). The experimental results are shown in Table 8. The results show that our PLA outperforms these transferable attack methods. We believe this is because PLA is specifically designed to trigger fine-tuned models to produce predetermined outputs, which can be understood as "fine-tuning transferability." In contrast, these methods focus on cross-model (cross-architecture or cross-prompt) transferability.

## D.4 TRACKING RESULTS WITH ADDITIONAL ORIGINAL MODELS

We also conduct experiments using QWEN2-VL-7B (Wang et al., 2024a) and InternVL2-2B (Chen et al., 2023) as the original models. The results are shown in Table 9. The experimental results demonstrate that our method is effective in protecting the copyright of QWEN2VL and InternVL2, further showing the generalizability of PLA to other LVLMs.

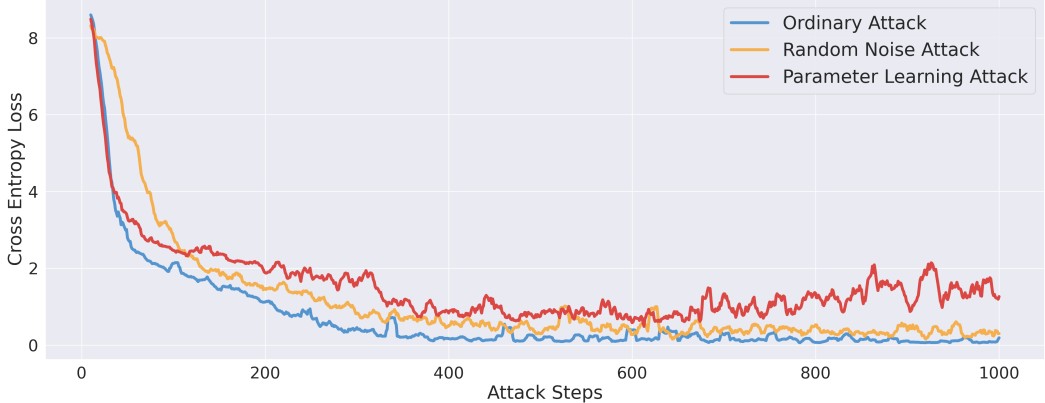

Figure 12: The loss decline of different attack methods.

Table 6: Copyright tracking results for models fine-tuned on additional downstream tasks.

| Method | RefCOCO-LoRA | RefCOCO-Full | HM-LoRA | HM-Full |
|---|---|---|---|---|
| Ordinary | 3% | 1% | 7% | 5% |
| IF | 16% | 12% | 22% | 24% |
| PLA(Ours) | 45% | 41% | 62% | 52% |

# E  ADDITIONAL ABLATION STUDIES

## E.1  ABLATION OF TRAINABLE MODULES IN FINE-TUNING

In the fine-tuning experiments, we set the trainable components to the MLP projector and the LLM by default, while keeping the vision encoder frozen, which is consistent with the instruction-tuning phase of LLaVA. We conduct ablation experiments on the trainable modules using the ChEBI-20 dataset, and the results are shown in Table 10. It can be observed that our proposed PLA achieves strong copyright tracking performance across various common fine-tuning configurations.

## E.2  ABLATION OF THE PERTURBATION BUDGET

The ablation results of the perturbation budget in trigger construction are shown in Table 11. Experimental results show that the tracking performance does not significantly improve when the perturbation budget exceeds 16/255. Therefore, considering the concealment of the triggers, we chose a perturbation budget of 16/255.

## E.3  ABLATION OF ATTACK STEPS

The ablation results of attack steps in trigger construction are shown in Table 12. Experimental results show that performance is poor when the number of attack steps is small; as the attack steps approach 1000, performance improves and begins to stabilize.

Table 7: Copyright tracking results on unrelated LVLMs.

| LVLMs | MiniGPT-4 | QWEN2-VL | InternVL2 | LLaVA-NEXT | InstructBLIP |
|---|---|---|---|---|---|
| PLA(Ours) | 0% | 0% | 0% | 0% | 0% |

Table 8: Comparison of our proposed method PLA with common transferable attack methods on the copyright tracking performance of fine-tuned models across 6 datasets.

| Method | V7W | ST-VQA | TextVQA | PaintingF | MathV | ChEBI | Average |
|---|---|---|---|---|---|---|---|
| Ordinary | 2% | 1% | 4% | 2% | 0% | 2% | 2% |
| MIM | 5% | 2% | 7% | 3% | 4% | 2% | 4% |
| DIM | 5% | 4% | 6% | 5% | 9% | 3% | 5% |
| CroPA | 3% | 1% | 5% | 3% | 0% | 2% | 2% |
| PLA (Ours) | **49%** | **58%** | **49%** | **63%** | **36%** | **56%** | **52%** |

Table 9: Tracking results of our proposed method PLA with additional original LVLMs.

| Original LVLM | Method | ST-VQA | PaintingF | MathV | ChEBI |
|---|---|---|---|---|---|
| InternVL2-2B | Ordinary | 3% | 3% | 1% | 4% |
| | PLA(Ours) | 45% | 57% | 36% | 42% |
| QWEN2-VL-7B | Ordinary | 1% | 2% | 2% | 3% |
| | PLA(Ours) | 51% | 65% | 47% | 59% |

Table 10: Ablation results of trainable modules with a single QA pair "*Q: Detecting copyright. A: ICLR Conference.*" on ChEBI-20 dataset.

| Trainable Modules | Projector+LLM | Vision Encoder+Projector+LLM | LLM | Projector |
|---|---|---|---|---|
| Ordinary | 3% | 4% | 8% | 6% |
| PLA(Ours) | 58% | 53% | 55% | 62% |

Table 11: Ablation results of the perturbation budget with a single QA pair "*Q: Detecting copyright. A: ICLR Conference.*" on ChEBI-20 dataset.

| Budget | 1 | 2 | 4 | 8 | 16 | 32 | 64 | 128 |
|---|---|---|---|---|---|---|---|---|
| PLA(Ours) | 0% | 0% | 0% | 19% | 58% | 63% | 52% | 59% |

Table 12: Ablation results of attack steps with a single QA pair "*Q: Detecting copyright. A: ICLR Conference.*" on ChEBI-20 dataset.

| Steps | 100 | 200 | 300 | 400 | 500 | 600 | 700 | 800 | 900 | 1000 | 1200 |
|---|---|---|---|---|---|---|---|---|---|---|---|
| PLA(Ours) | 0% | 0% | 10% | 10% | 32% | 35% | 43% | 48% | 56% | 58% | 53% |

