# OpenReview forum: "Tracking the Copyright of Large Vision-Language Models through Parameter Learning Adversarial Images"
_ICLR.cc/2025/Conference — ICLR 2025 Poster_

### Official Review · Reviewer_PhEj · 2024-11-02

**Soundness:** 3
**Presentation:** 3
**Contribution:** 3
**Rating:** 6
**Confidence:** 3

**Summary:**

The paper propose a method PLA to track copyrights and detect unlicensed usage of LVLMs. The method is to design rare question-answer pairs, and optimize corresponding adversarial images. These adv image-text pairs are used as triggers to detect if the model are copyright infringing. To increase generalization, the paper adds gradient-based updates to model parameters when optimizing adv images.

**Strengths:**

1) The topic is both significant and practical.

2) The writing is clear and easy to follow.

3) The method is novel and the experiments show a clear increase compared to previous ones.

**Weaknesses:**

Please refer to the questions.

minor: line 94: traqcking

**Questions:**

1) The experimental results include finetuning on Llava1.5, ST-VQA, etc. Have the authors tried some other advanced models such as miniGPT4, QWEN2-VL (2b or 7b version), and InternVL (2b version )? What are the results?

2) The paper finetunes the model on certain datasets using both full finetuning and lora. Can the authors provide the performances on relevant benchmarks of the VLMs before and after fine-tuning? This is to check if the fine-tuning is conducted correctly.

3) can the authors also provide the TMR for related and unrelated models? For example, if the image-text pairs are designed for llava1.5, what's the TMR for llava1.6 and for miniGPT4?

I'm willing to raise the score once the questions are addressed.

---

> ### Author Response · Authors · 2024-11-22
> **Response to Reviewer PhEj**
>
> We sincerely thank you for acknowledging the novelty of our method and for your positive feedback. We would like to address the questions individually below:
>
> > W'1
>
> Thank you for pointing out the spelling error. We will correct it in the revised version.
>
> > Q'1 Results of other advanced LVLMs
>
> Based on your suggestion, we conducted experiments using QWEN2-VL-7B and InternVL2-2B as the original models. Since the fine-tuning code for MiniGPT-4 is not open-sourced, we opted to use these two models within the limited rebuttal period. The results are shown in the table below.
>
> |           | **Original LVLM** | **ST-VQA** | **PaintingF** | **MathV** | **ChEBI** |
> | --------- | ----------------- | ---------- | ------------- | --------- | --------- |
> | Ordinary  | InternVL2-2B      | 3%         | 3%            | 1%        | 4%        |
> |           | QWEN2-VL-7B       | 1%         | 2%            | 2%        | 3%        |
> | PLA(Ours) | InternVL2-2B      | 45%        | 57%           | 36%       | 42%       |
> |           | QWEN2-VL-7B       | 51%        | 65%           | 47%       | 59%       |
>
> The experimental results demonstrate that our method is effective in protecting the copyright of QWEN2VL-7B and InternVL2-2B, further showing the generalizability of PLA to other LVLMs.
>
> > Q'2 Performance of benchmarks after fine-tuning
>
> We tested LLaVA’s performance on benchmarks related to several specific datasets before and after fine-tuning, and the results show that fine-tuning was conducted as expected, as shown in the table below.
>
> | **Benchmarks**         | **STVQA (Acc)** | **TextVQA (Acc)** | **PaintingForm (BLEU)** | **MathV (Acc)** | **ChEBI (BLEU)** |
> | ---------------------- | --------------- | ----------------- | ----------------------- | --------------- | ---------------- |
> | Before fine-tuning     | 38.1            | 57.6              | 9.9                     | 10.2            | 15.9             |
> | After full fine-tuning | 46.4            | 62.1              | 18.7                    | 12.6            | 27.4             |
>
> > Q'3 Tracking performance on unrelated LVLMs
>
> PLA constructs triggers by simulating fine-tuning behavior through parameter reverse updates, which does not incorporate any design to enable the cross-model & cross-architecture transferability of triggers. As a result, the triggers do not cause unrelated models to produce the target output. The tracking results for models unrelated to LLaVA demonstrate that triggers generated by PLA do not falsely track these models, as shown in the table below.
>
> | **LVLMs** | **MiniGPT-4** | **QWEN2-VL** | **InternVL2** | **LLaVA1.6** | **InstructBLIP** |
> | --------- | ------------- | ------------ | ------------- | ------------ | ---------------- |
> | PLA       | 0%            | 0%           | 0%            | 0%           | 0%               |
>
> It is worth noting that LLaVA 1.6 (LLaVA-NEXT) was not obtained by fine-tuning LLaVA 1.5. Instead, it utilized a two-stage training process and was not trained based on LLaVA 1.5. In our paper, related models refer to those obtained through fine-tuning the original model. Therefore, the trigger tracking success rate for LLaVA 1.6 is 0%, as it is an unrelated model.
>
>
> We appreciate your feedback and hope our response addresses your points. We are open to further discussion if needed. Thank you once again for recognizing our work.

---

> > ### Comment · Reviewer_PhEj · 2024-12-01
> >
> > I encourage the authors to add these results to the final version, as they can address the readers' questions. The results added during the rebuttal period are pretty good, and better demonstrate the effectiveness.

---

> ### Author Response · Authors · 2024-11-26
> **We hope that our response addresses your questions**
>
> Dear Reviewer PhEj,
>
> We greatly appreciate the time you've invested in reviewing our response. Having submitted our rebuttal, we are eager to know if our response has addressed your questions. The discussion phase is nearing its end, but we have not yet received your response. We look forward to hearing from you for any further clarification that you might require.
>
> Best,
>
> Submission 13438 Authors

---

> ### Author Response · Authors · 2024-11-30
> **Gentle reminder of the author-reviewer discussion deadline**
>
> Dear Reviewer PhEj,
>
> We sincerely thank you for your time and effort in reviewing our paper and for your thoughtful feedback and appreciation of this work! We have carefully read and addressed your questions. As the discussion period deadline approaches (Dec.3, 2024), if you have any additional questions, we would be delighted to continue the conversation with you! We look forward to hearing from you for any further clarification that you might require.
>
> Best,
>
> Submission 13438 Authors

---

> ### Author Response · Authors · 2024-12-02
>
> Dear Reviewer PhEj,
>
> Thank you for your encouraging feedback and for taking the time to review our rebuttal. We are delighted to hear that you find the results added during the rebuttal period valuable and better demonstrating the effectiveness of our approach.
>
> We have added the experimental results addressing Q1 and Q3 in Sections D.4 and D.2 of the revised paper's appendix before. Since the deadline for submitting the revised PDF has passed, we are unable to re-upload a new revision containing the results for Q2. Nonetheless, we are prepared to include the experimental results for Q2 in the final version.
>
> Building on your initial positive assessment of this work, we hope these results and improvements will further reinforce your recommendation for acceptance. Thank you once again for your constructive review and support in improving our work!
>
> Best,
>
> 13438 Authors

---

### Official Review · Reviewer_Rr8h · 2024-11-03

**Soundness:** 3
**Presentation:** 3
**Contribution:** 3
**Rating:** 6
**Confidence:** 4

**Summary:**

The paper focuses on identifying whether a model has been fine-tuned based on a publicly released pre-trained visual-language model.
It proposes the use of adversarial data, generated through targeted attacks on the original pre-trained model, to detect outputs from downstream models, thereby assessing potential copyright issues.
To enhance generalization, the paper introduces a Parameter Learning Attack (PLA), which incorporates an adversarial training process to simulate parameter changes in downstream tasks.
Experiments on six downstream VQA datasets show that adversarial data generated by PLA can effectively track the copyright of pre-trained VLMs, achieving satisfactory accuracy.

**Strengths:**

1. The issue addressed in this paper—copyright tracking for open-source VLMs—is highly significant. The approach of constructing adversarial examples to detect outputs from downstream models provides an insightful solution for identifying unlicensed usage of pre-trained VLMs.
2. The paper is well-structured and easy to follow.

**Weaknesses:**

1. Although the paper presents a novel scenario, the techniques employed to address the problem are not particularly innovative. The core issue can be understood as constructing adversarial examples with cross-model transferability based on a given pre-trained model through targeted adversarial attacks. This is a widely studied topic in the field of adversarial attacks. While the application of targeted attacks for copyright tracking is indeed creative, the paper overlooks many comparative methods, such as [1]-[5]. I believe that most adversarial attacks focusing on cross-model transferability could be tailored for the task presented in this paper.

[1] Dong et al. Boosting Adversarial Attacks with Momentum.
[2] Xie et al. Improving Transferability of Adversarial Examples with Input Diversity.
[3] Lu et al. Set-level Guidance Attack: Boosting Adversarial Transferability of Vision-Language Pre-training Models.
[4] Luo et al. An Image Is Worth 1000 Lies: Transferability of Adversarial Images across Prompts on Vision-Language Models.
[5] Yin et al. VQAttack: Transferable Adversarial Attacks on Visual Question Answering via Pre-trained Models.

2. The paper should evaluate the performance of PLA under varying degrees of model parameter changes. The experiments conducted are limited to relatively simple VQA datasets, where the parameter changes are minimal. If fine-tuning occurs on more complex datasets (such as ScienceQA), the extent of parameter changes may be more substantial. How would PLA perform under such conditions?

**Questions:**

See in Weaknesses.

---

> ### Author Response · Authors · 2024-11-22
> **Response to Reviewer Rr8h**
>
> Thank you for your thorough reading of our paper. In response to your concerns, we provide the following analysis.
>
> > W'1 Cross-model transferable adversarial attacks
>
> We believe there is a misunderstanding regarding your perspective. We want to clarify that **cross-model transferable adversarial attacks and adversarial example-based copyright tracking are fundamentally different.** The former refers to crafting adversarial examples on a specific model that can also effectively attack models with different architectures performing the same task. In contrast, the latter involves crafting adversarial examples on an original model that can only effectively attack models fine-tuned from that original model, while not attacking other models (e.g., models with different architectures).
>
> **Therefore, adversarial examples with cross-model transferability are not designed for copyright tracking.** Copyright tracking requires that only related models, fine-tuned from the original model (LLaVA), are triggered to produce specific outputs. Cross-model transferable adversarial examples might cause unrelated LVLMs (e.g., QWEN2-VL, InternVL2) to also produce specific outputs. This could result in false tracking and lead to overclaims. Therefore, we did not compare our method with transferable adversarial attack methods.
>
> To validate the effectiveness of our proposed PLA, we compared it with some of the cross-model transferable attack methods you mentioned. The results are as follows. Note that the SGA and VQAttack you mentioned are designed for Vision-Language Pretraining (VLP) models rather than Large Vision-Language Models (LVLMs).
>
> |           | **V7W** | **ST-VQA** | **TextVQA** | **PaintingF** | **MathV** | **ChEBI** | **Average** |
> | --------- | ------- | ---------- | ----------- | ------------- | --------- | --------- | ----------- |
> | Ordinary  | 2%      | 1%         | 4%          | 2%            | 0%        | 2%        | 2%          |
> | MIM [1]   | 5%      | 2%         | 7%          | 3%            | 4%        | 2%        | 4%          |
> | DIM [2]   | 5%      | 4%         | 6%          | 5%            | 9%        | 3%        | 5%          |
> | CroPA [3] | 3%      | 1%         | 5%          | 3%            | 0%        | 2%        | 2%          |
> | PLA(Ours) | **49%** | **58%**    | **49%**     | **63%**       | **36%**   | **56%**   | **52%**     |
>
> The results show that our PLA outperforms these transferable attack methods. We believe this is because PLA is specifically designed to trigger fine-tuned models to produce predetermined outputs, which can be understood as “fine-tuning transferability.” In contrast, these methods focus on cross-model (cross-architecture or cross-prompt) transferability.
>
> > W'2 Complex datasets
>
> We would like to clarify that we used two domain-specific downstream datasets in our main experiments: a chemistry molecular description dataset ChEBI-20 and a mathematical image question-answering dataset MathV360K, which involve tasks in the fields of chemistry and mathematics. Sample examples can be found in Appendix Section B. We believe these datasets are not “simple” tasks for LLaVA. Firstly, the image data distribution in these datasets differs significantly from that in LLaVA’s pretraining. Secondly, the task-specific knowledge required for these datasets also deviates from LLaVA’s pretrained knowledge. Both datasets require the model to adapt to highly specialized knowledge. Nevertheless, PLA still demonstrates strong tracking performance on models fine-tuned from these tasks.
>
> [1] Dong et al. Boosting Adversarial Attacks with Momentum.
>
> [2] Xie et al. Improving Transferability of Adversarial Examples with Input Diversity.
>
> [3] Luo et al. An Image Is Worth 1000 Lies: Transferability of Adversarial Images across Prompts on Vision-Language Models.
>
>
> We hope our response addresses your concerns. If you have any further questions, please feel free to reach out.

---

> > ### Comment · Reviewer_Rr8h · 2024-11-27
> > **Response to the Authors**
> >
> > Thanks for your explanations. I have updated my rating to 6.

---

> ### Author Response · Authors · 2024-11-26
> **We hope that our response addresses your questions**
>
> Dear Reviewer Rr8h,
>
> We greatly appreciate the time you've invested in reviewing our response. Having submitted our rebuttal, we are eager to know if our response has addressed your questions. The discussion phase is nearing its end, but we have not yet received your response. We look forward to hearing from you for any further clarification that you might require.
>
> Best,
>
> Submission 13438 Authors

---

### Official Review · Reviewer_BfWs · 2024-11-04

**Soundness:** 2
**Presentation:** 3
**Contribution:** 2
**Rating:** 5
**Confidence:** 4

**Summary:**

This paper proposes a method for tracking the copyright of LVLMs, i.e. detecting whether an LVLM is finetuned from another one. They use adversarial attacks to create trigger images that, when paired with specific questions, elicit predetermined responses from models derived from the original model. The trigger is created in an adversarial training fashion, where the model is fine tuned to resist the attack while the attack is created again on the robustified model. The authors evaluate their method on LLaVA-1.5 across various fine-tuning scenarios and datasets, demonstrating improved detection rates compared to baseline methods.

**Strengths:**

Problem significance. This paper addresses an important and timely issue in AI model protection.

Writing is generally clear and easy to read.

**Weaknesses:**

Methodological limitations.
1. The authors do not provide detailed description on which part of the LVLM is being fine tuned. It seems that they only consider language model fine-tuning and ignore vision encoder and adapter fine-tuning, which does not cover the full picture of fine-tuning. The authors should provide analyses on how finetuning different parts of the LVLM influences the detection rate.
1. In line 353, why do you need to enhance the concealment of trigger images? For copyright detection purposes, there is no inherent need for imperceptibility. Arbitrary patterns or even pure noise can work as triggers, since the goal is detection accuracy, not visual stealth. And the perturbation budget of ε=16/255 seems arbitrary, and the author should provide varying levels of perturbation budgets and reveal the trend of detection rate as the budget increases.

Insufficient experiments and analyses.
1. This paper does not provide analysis on how to determine the number of steps for adversarial attacks. The authors should provide the detection rate vs the number of attack steps.
1. There is no proper treatment of false positives and false negatives. And it is missing ROC curves and threshold analysis. Not clear how this method works in practice.
1. Why would model creators prefer post-release triggers than finger-prints implemented during training? This paper shows in table 1 that the proposed method is more effective than training time triggers. It should elaborate more on this aspect by showing more comparisons.

**Questions:**

1. Can you show more details of how IF (Xu et al., 2024) is implemented in this paper? Why is IF less effective than the proposed method that does not embed trigger information into the model during training?

1. How does the method handle model ensemble or knowledge distillation?

1. Is this approach still effective when the model stealer finetunes the model using adaptive methods to evade detection?

---

> ### Author Response · Authors · 2024-11-22
> **Response to Reviewer BfWs (1/2)**
>
> We thank the reviewer for the thorough review of our work. We have taken note of the concerns raised by the reviewer and below we will address them accordingly.
>
> > W'1 Fine-tuning modules of LVLMs
>
> In the fine-tuning experiments, we set the trainable components to the MLP projector and the LLM by default, while keeping the vision encoder frozen, which is consistent with the instruction-tuning phase of LLaVA. We have now conducted ablation experiments on the trainable modules using the ChEBI dataset, and the results are shown below (full fine-tuning for LLM).
>
> | Trainable module(s) | Projector + LLM (default) | Vision Encoder + Projector + LLM | LLM  | Projector |
> | ------------------- | ------------------------- | -------------------------------- | ---- | --------- |
> | Ordinary            | 3%                        | 4%                               | 8%   | 6%        |
> | PLA(Ours)           | 58%                       | 53%                              | 55%  | 62%       |
>
> It can be observed that our proposed PLA achieves strong copyright tracking performance across various common fine-tuning configurations. Notably, PLA achieves the highest performance (62%) on projector-tuned models. This is likely because the smaller number of trainable parameters results in less significant changes to the original model. We have included these new experimental results in the revised version of our paper.
>
> > W'2 Perturbation budget
>
> Enhancing the concealment of trigger images can help prevent stealers from detecting the image input. If the perturbation budget is too large, stealers may identify anomalies in the input images, causing the model to refuse to respond and thereby evade copyright tracking. We have conducted an ablation study on the perturbation budget using the ChEBI dataset, and the results are shown below.
>
> | Budget | 1    | 2    | 4    | 8    | 16   | 32   | 64   | 128  |
> | ------ | ---- | ---- | ---- | ---- | ---- | ---- | ---- | ---- |
> | TMR    | 0%   | 0%   | 0%   | 19%  | 58%  | 63%  | 52%  | 59%  |
>
> Experimental results show that the tracking performance does not significantly improve when the perturbation budget exceeds 16/255. Therefore, considering the concealment of the triggers, we chose a perturbation budget of 16/255.
>
> > W'3 Attack steps
>
> We conducted an ablation study on the attack steps using the ChEBI dataset. The change of TMRs with the number of attack steps is shown below.
>
> | Steps | 100  | 200  | 300  | 400  | 500  | 600  | 700  | 800  | 900  | 1000 | 1200 |
> | ----- | ---- | ---- | ---- | ---- | ---- | ---- | ---- | ---- | ---- | ---- | ---- |
> | TMR   | 0%   | 0%   | 10%  | 10%  | 32%  | 35%  | 43%  | 48%  | 56%  | 58%  | 53%  |
>
> Experimental results show that performance is poor when the number of attack steps is small; as the attack steps approach 1000, performance improves and begins to stabilize.
>
> > W'4 False positives and false negatives
>
> PLA constructs triggers by simulating fine-tuning behavior through parameter reverse updates, which does not incorporate any design to enable the cross-model & cross-architecture transferability of triggers. As a result, the triggers do not cause unrelated models to produce the target output. The tracking results for models unrelated to LLaVA demonstrate that triggers generated by PLA do not falsely track these models, as shown in the table below.
>
> | **LVLMs** | **MiniGPT-4** | **QWEN2-VL** | **InternVL2** | **LLaVA1.6** | **InstructBLIP** |
> | --------- | ------------- | ------------ | ------------- | ------------ | ---------------- |
> | PLA       | 0%            | 0%           | 0%            | 0%           | 0%               |

---

> > ### Author Response · Authors · 2024-11-22
> > **Response to Reviewer BfWs (2/2)**
> >
> > > W'5 & Q'1 PLA vs IF
> >
> > **Why prefer post-release triggers**
> >
> > Because the proposed triggers are non-intrusive, they do not alter the parameters of the released model or affect its performance and behavior, thus avoiding potential harm to the protected model.
> >
> > **Details of IF**
> >
> > In line 365, we reported the implementation details of IP. Specifically, we designed textual question-answer pairs consistent with IF and used completely black images (with all pixel values set to 0) to construct the fingerprints. These fingerprint samples were then combined with benign instruction-tuning data at a 1:5 ratio, as described in the paper of IF, to fine-tune LLaVA for memory implantation.
> >
> > **Reason for the performance difference**
> >
> > Experimental results demonstrate that PLA outperforms IF, which may be attributed to two main reasons.
> >
> > First, the IF method was originally designed to track the copyright of LLMs, but the architecture of LVLMs differs significantly from that of LLMs. The introduction of the visual modality may render IF ineffective.
> >
> > Second, IF, as an invasive method, implants memory by altering a small number of model parameters. However, these parameters may be overwritten during downstream fine-tuning. In contrast, PLA generates triggers by simulating fine-tuning behavior without modifying the published model parameters, making it more robust to downstream fine-tuning.
> >
> > > Q'2 Model ensemble and knowledge distillation
> >
> > Since our proposed PLA relies on model parameters to construct triggers, it is methodologically unable to handle cases where stealers employ model ensembling or knowledge distillation. We plan to explore this limitation in future work. Notably, our work primarily focuses on tracking the copyright of fine-tuned models, and related works [1] [2] also typically do not consider model ensembling or knowledge distillation.
> >
> > > Q'3 Adaptive fine-tuning methods
> >
> > What exactly do adaptive methods refer to? We believe that our proposed PLA could be effective for certain adaptive fine-tuning methods, such as fine-tuning with partial random initialization of model parameters or fine-tuning after model pruning. For these adaptive methods, PLA’s performance depends on the extent of random initialization or pruning before tuning. In Section 4.3 of the paper, we report PLA’s robustness to model pruning during testing phase, and the results show that PLA still works when pruning size is 10%. Similarly, PLA may also exhibit robustness to the mentioned adaptive fine-tuning methods.
> >
> > [1] Xu et al. Instructional Fingerprinting of Large Language Models. 2024.
> > [2] Li et al. Double-i Watermark: Protecting Model Copyright for LLM Fine-tuning. 2024.
> >
> > We hope this response clarifies your concerns. If there are any further inquiries, please let us know. We are happy to engage in further discussion.

---

> ### Author Response · Authors · 2024-11-26
> **We hope that our response addresses your concerns**
>
> Dear Reviewer BfWs,
>
> We greatly appreciate the time you've invested in reviewing our response. Having submitted our rebuttal, we are eager to know if our response has addressed your concern. The discussion phase is nearing its end, but we have not yet received your response. We look forward to hearing from you for any further clarification that you might require.
>
> Best,
>
> Submission 13438 Authors

---

> ### Author Response · Authors · 2024-11-30
> **Gentle reminder of the author-reviewer discussion deadline**
>
> Dear Reviewer BfWs,
>
> We greatly appreciate your time and effort in reviewing our work! In our previous responses, we have actively addressed your concerns by supplementing the relevant experiments you mentioned and providing detailed explanations and clarifications regarding your questions about the paper.
>
> As the discussion period deadline approaches (Dec.3, 2024), if you have any additional questions, we would be delighted to continue the conversation with you! We look forward to hearing from you for any further clarification that you might require.
>
> Best,
>
> Submission 13438 Authors

---

> ### Comment · Reviewer_BfWs · 2024-12-03
>
> Thank you for your response! After reading it, I still have a question regarding the motivation and influence of concealment. In principle the input image can be arbitrary, how would anomaly detection work in this scenario? This raises another question that whether the "rare question-answer pairs" used for tracking will be detected as anomaly in a similar manner?
>
> From the results of varying the budget, the TMR will decrease after the budget exceeds 32, this is counter-intuitive since a larger budget should strictly outperform lower budgets if the optimization is correct.

---

> > ### Author Response · Authors · 2024-12-03
> >
> > Thank you for your comments!
> >
> > Regarding your questions, we offer the following explanation. While the input images are arbitrary, certain techniques can detect potential adversarial images, such as frequency domain analysis or local variance detection in the pixel space. A large perturbation budget may make the trigger images more easily detectable through similar pixel-level analyses. For textual inputs, the question-answer pairs we designed do not include malicious phrases or nonsensical strings, making them less likely to be flagged as abnormal.
> >
> > As for the results of the budget experiments, we believe the performance degradation observed with budgets exceeding 32 is due to the increased learnable space in adversarial images, which may lead to overfitting on the original model. This, in turn, reduces their success rate in triggering fine-tuned models. Overall, we consider selecting a perturbation budget of 16 to be a more reasonable choice.
> >
> > Thank you again for your feedback, and we hope our explanation addresses your concerns.

---

### Official Review · Reviewer_gTwA · 2024-11-07

**Soundness:** 4
**Presentation:** 3
**Contribution:** 4
**Rating:** 8
**Confidence:** 4

**Summary:**

This paper tackles the problem of unauthorized usage and copyright infringement of LVLMs due to unlicensed fine-tuning of publicly available models. It proposes a method called Parameter Learning Attack (PLA) to address this issue by enabling copyright tracking without modifying the original model. PLA employs an adversarial attack to generate trigger images that produce distinct outputs for copyright verification. To enhance robustness, it involves adjustment of model parameters, ensuring the trigger images remain effective even on fine-tuned versions. This approach is non-intrusive, as it can be applied post-release without impacting the model’s performance. Experiments simulating real-world fine-tuning scenarios, show that PLA outperforms baseline methods in accurately identifying original ownership. It offers a solution for detecting unauthorized LVLM usage.

**Strengths:**

The proposed approach PLA offers several valuable strengths:
- is a non-intrusive way of copyright tracking
- generates robust trigger images that are resistant to model finetuning
- outperforms traditional methods by reliably identifying copyright ownership

The paper is well-written, presenting concepts clearly and systematically.

**Weaknesses:**

- PLA’s core innovation involves updating parameters in reverse to increase trigger robustness; however, the paper lacks an in-depth explanation of why this specific approach effectively simulates fine-tuning.
- The experiments focus solely on fine-tuning for VQA tasks, which limits the assessment of PLA’s robustness across broader vision-language tasks. Testing on additional tasks, such as image captioning, visual grounding, or multi-modal classification, would provide a more comprehensive evaluation of PLA's effectiveness in diverse VL applications.
- Scalability and Computational Efficiency: Reverse parameter updates during adversarial training may be computationally intensive, raising questions about PLA’s scalability for large models. Adding discussion or experiments on PLA’s computational demands, along with potential optimizations, would improve its applicability for real-world use.
- Limited novelty:  While the paper proposes PLA for copyright tracking, the use of adversarial triggers to mark models has similarities to prior watermarking and adversarial attack research.

**Questions:**

- Is the statement that "changing the weights in the attention layers has a greater impact on trigger images than altering the weights in the MLP layers" based on empirical observation? Could you provide further reasoning or experimental data to support this claim?
- Have you considered potential vulnerabilities where attackers might detect or circumvent PLA’s adversarial triggers? If so, what measures could be implemented to defend against such countermeasures?
- How does PLA’s approach of reverse parameter updates specifically differentiate itself from existing model watermarking and adversarial trigger methods?

---

> ### Author Response · Authors · 2024-11-22
> **Response to Reviewer gTwA (1/2)**
>
> We sincerely appreciate your acknowledgment of our work and your recognition of our method. Regarding your concerns, we hope that our clarification can be helpful.
>
> > W'1 Lack of explanation
>
> Model fine-tuning adapts the model to downstream tasks through parameter updates, and these updates inherently alter the model’s behavior, making traditional adversarial examples ineffective in triggering fine-tuned models. The core idea of PLA is to make triggers adapt to the model’s behavioral changes by simulating the behavior of a fine-tuned model rejecting the target output. During copyright tracking, since the triggers have already converged under the resistance of parameter updates (model rejection), they are more likely to successfully trigger the actual fine-tuned model to produce the predetermined output.
>
> > W'2 Broader vision-language Tasks
>
> Actually, the PaintingForm and ChEBI datasets we utilized belong to image captioning tasks, with relevant examples provided in Section B in the Appendix. We have now fine-tuned LLaVA on visual grounding and multimodal classification tasks. And then we evaluated the trigger’s performance to track the copyright of fine-tuned models. The results are shown in the table below.
>
> |           | RefCOCO (LoRA) | RefCOCO (Full) | HatefulMemes (LoRA) | HatefulMemes (Full) |
> | --------- | -------------- | -------------- | ------------------- | ------------------- |
> | Ordinary  | 3%             | 1%             | 7%                  | 5%                  |
> | IF        | 16%            | 12%            | 22%                 | 24%                 |
> | PLA(Ours) | 45%            | 41%            | 62%                 | 52%                 |
>
> Specifically, the RefCOCO dataset is used for the visual grounding task, while the Hateful Memes dataset is used for the multimodal classification task. The results indicate that our method remains effective in these tasks, achieving better performance compared to baseline method IP. Thanks for raising the point about broader vision-language tasks. We have included these new experimental results in the revised manuscript.
>
> > W'3 Scalability and computational efficiency
>
> **Computational efficiency**
>
> While PLA introduces reverse parameter updates, these updates are confined to the adversarial trigger generation phase and are computationally less demanding compared to full model fine-tuning. Specifically, we employed a simple gradient ascent algorithm to perform reverse updates on the model parameters, and the number of triggers was significantly smaller than the actual training data. Thus, PLA is more efficient compared to fine-tuning. Additionally, adversarial sample generation is a one-time process, so PLA is computationally feasible for large-scale applications.
>
> **Potential optimizations**
>
> To further improve efficiency, we believe that LoRA can be utilized for reverse updates of the model in PLA. Besides, the efficiency of PLA can be enhanced by reducing the number of attack steps. However, there is a trade-off between efficiency and performance, as decreasing the number of iterations may lead to performance decline.
>
> > W'4 & Q'3 Novelty of our work
>
> We highlight the novelty of our work from the following perspectives:
>
> - **Novel Direction**: We are the first to address copyright protection for LVLMs, which is an unexplored area.
> - **Novel Motivation**: Unlike prior watermarking methods that commonly use backdoor attacks for fingerprint embedding, we employ adversarial attacks to track copyrights, offering a non-intrusive approach. Furthermore, existing adversarial attack studies related to LVLMs have not focused on the copyright protection task.
> - **Novel Approach**: Leveraging the characteristics of LVLMs, we innovatively design adversarial triggers capable of tracking the original copyright of fine-tuned models.

---

> > ### Author Response · Authors · 2024-11-22
> > **Response to Reviewer gTwA (2/2)**
> >
> > > Q'1 Observation in robustness analysis
> >
> > The statement that "changing the weights in the attention layers has a greater impact on trigger images than altering the weights in the MLP layers" in our paper is based on empirical observations, as demonstrated in our robustness experiments. To further validate this conclusion, we have conducted additional experiments for model perturbation by varying the perturbation sizes of Attention and MLP layers while keeping the total noise threshold fixed (10% in total). The results are shown below:
> >
> > |              | No Perturbation | A. 2% + M. 8% | A. 4% + M. 6% | A. 6% + M. 4% | A. 8% + M. 2% |
> > | ------------ | --------------- | ------------- | ------------- | ------------- | ------------- |
> > | ST-VQA       | 53%             | 49%           | 50%           | 47%           | 44%           |
> > | PaintingForm | 71%             | 70%           | 68%           | 68%           | 64%           |
> > | ChEBI        | 58%             | 56%           | 57%           | 53%           | 50%           |
> >
> > A. refers to Attention module and B. refers to MLP module. We can observe that perturbing Attention layers with Gaussian noise cause a larger performance drop compared to perturbing MLP layers. Nevertheless, we still do not fully understand the underlying reasons behind this phenomenon, which we plan to explore in future work.
> >
> > > Q'2 Measures against detection by stealers
> >
> > Yes, stealers might detect input triggers and evade copyright tracking. To address this, we used a small adversarial perturbation magnitude (16/255) to avoid detection on input images, as larger perturbations could make the noise too noticeable. Additionally, we employed conventional trigger questions to prevent the detection of input text. In contrast, some works [1] use meaningless textual strings as questions, which can be easily detected by stealers.
> >
> > [1] Xu et al. Instructional Fingerprinting of Large Language Models. 2024.
> >
> > We hope the response addresses your concerns. If you have any further inquiries, please feel free to reach out. We are open to continued discussion.

---

> ### Author Response · Authors · 2024-11-26
> **We hope that our response addresses your concerns**
>
> Dear Reviewer gTwA,
>
> We greatly appreciate the time you've invested in reviewing our response. Having submitted our rebuttal, we are eager to know if our response has addressed your concern. The discussion phase is nearing its end, but we have not yet received your response. We look forward to hearing from you for any further clarification that you might require.
>
> Best,
>
> Submission 13438 Authors

---

> > ### Comment · Reviewer_gTwA · 2024-11-28
> >
> > Thanks for the rebuttal. It addresses most of my concerns. I will revisit my ratings.

---

### Author Response · Authors · 2024-11-22
**General Response**

Dear reviewers,

We want to thank you for your thorough and detailed reviews. We believe your suggestions have improved our paper. Considering all the reviewers’ comments, we have provided individual responses to each reviewer. Furthermore, we have added additional experiments and analysis in the paper's appendix.  In the following, we summarize the changes made to the revision which are marked in blue.

- Appendix D.1: Added tracking results for fine-tuned models on additional tasks. (@Reviewer gTwA)
- Appendix D.2: Evaluated tracking performance on unrelated LVLMs. (@Reviewer BfWs, PhEj)
- Appendix D.3: Comparison with transferable attack methods. (@Reviewer Rr8h)
- Appendix D.4: Tracking results using additional original LVLMs . (@Reviewer PhEj)
- Appendix E.1: Ablation results of trainable modules in fine-tuning. (@Reviewer BfWs)
- Appendix E.2: Ablation results of the perturbation budget. (@Reviewer BfWs)
- Appendix E.3: Ablation results of attack steps. (@Reviewer BfWs)

Your insights have been invaluable in enhancing the overall quality of our work.

Sincerely,

Authors of Submission 13438.

---

### Meta-Review · Area_Chair_9N1s · 2024-12-21

**Metareview:**

1x accept, 2x borderline accept, 1x borderline reject. This paper introduces a Parameter Learning Attack method to track copyrights in large vision-language models by generating robust adversarial triggers that remain effective even after fine-tuning. The reviewers agree on the (1) practical significance of addressing unauthorized fine-tuning, (2) clear experiments and presentation of the proposed adversarial trigger design, and (3) notable empirical results demonstrating consistent detection of derived/fine-tuned models. However, they note (1) limited evaluation on more advanced fine-tuning scenarios and tasks, (2) insufficient comparisons to broader cross-model adversarial/watermarking baselines, and (3) a need for deeper discussion on concealment vs. perturbation budget and potential evasion strategies. The authors have followed up with extensive clarifications and new experiments on different downstream tasks (e.g., RefCOCO, HatefulMemes), ablation studies on training modules, and comparisons to cross-model adversarial attacks, so the AC leans to accept this submission.

**Additional Comments On Reviewer Discussion:**

N/A

---

### Decision · Program_Chairs · 2025-01-22

Accept (Poster)